# Multifunctional Metallic Nanowires in Advanced Building Applications

**DOI:** 10.3390/ma12111731

**Published:** 2019-05-28

**Authors:** Kwok Wei Shah, Teng Xiong

**Affiliations:** Department of Building, School of Design and Environment, National University of Singapore, 4 Architecture Drive, Singapore 117566, Singapore; xiongteng@u.nus.edu

**Keywords:** metallic nanowires, nanotechnology, building applications, synthesis, characterization

## Abstract

Metallic nanowires (NWs) have attracted great attention in the frontiers of nanomaterial science due to their extraordinary properties, such as high thermal and electrical conductivity, high aspect ratio, good mechanical flexibility, and excellent optical transparency. The metallic NWs and their nanocomposites, as a promising alternative for conventional building materials, have been extensively studied recently, but review works on these novel versatile nanostructures and their various uses in the building and construction industry are still lacking. We present a comprehensive review on current state-of-the-art research and progress regarding multifunctional metallic NWs and their specific building applications, including thermal energy storage (TES), thermal transport, electrochromic windows (ECW), as well as photovoltaic (PV) cells. The nanosynthesis techniques and nanocharacterization of silver nanowires (AgNWs) and copper nanowires (CuNWs) are overviewed and compared with each other. In addition, the fundamentals of different NWs for advanced building applications are introduced. Further discussion is presented on the improved performance of base materials by using these nanostructures, highlighting the key factors exhibiting their superior performance. Finally, the key benefits and limitations of metallic NWs for new generation building materials are obtained.

## 1. Introduction

The great advances achieved in nanotechnology play a key role in many modern industries. The fundamental properties of a material can dramatically change when its dimensions are reduced to the nanoscale, which can be achieved by physical or chemical methods [1]. As a useful one-dimensional (1D) nanostructure, metallic nanowires (NWs) have been extensively investigated for their unique properties, such as good thermal and electrical conductivity, high aspect ratio, good mechanical flexibility, excellent optical transparency, etc. On this basis, metallic NWs are being considered as multifunctional engineering materials owing to their amazing behaviors, which are not seen in their bulk counterparts [2,3]. In the past two decades, research into 1D metallic NWs has encouraged breakthrough technologies in a variety of fields, including flexible transparent conductive films and electrodes [4,5,6], conductive polymer nanocomposites [7], touch sensors [8,9], photovoltaic (PV) cells [10,11], electrochromic devices (ECD) [12], wearable optoelectronic devices [13], and thermal energy storage (TES) [14].

With the rapid urbanization and improvement of people’s living standard, the strong demand for sustainable and energy-efficient building design has become a pressing issue. Typical problems of the current building industry include high energy consumption and poor indoor thermal comfort [15]. In this context, the building industry is searching for a novel and cost-effective approach to improve conventional building materials by using various manufactured nanomaterials (MNWs), which has opened up new scientific opportunities for metallic NWs and their nanocomposites [16]. The metallic NW technology not only improves the functionalities of base materials (e.g., thermal conductivity [17] and flexibility [18]) but also endows highly valuable properties (e.g., hydrophobicity [19] and self-cleaning [20,21]), enabling ground-breaking building applications ranging from energy efficiency to indoor environment quality (IEQ).

The outstanding potentials of metallic NWs to facilitate energy-efficient building design, as well as a comfortable built environment, stimulate researchers in the field to investigate and fill the missing gaps in current knowledge and techniques. The rapid advancement in the frontier of metallic NW technologies, particularly silver nanowires (AgNWs) and copper nanowires (CuNWs), has made them up-and-coming candidates for next generation building materials. Many synthesis methods have been developed to yield AgNWs and CuNWs with uniform size at low cost thanks to the developed characterization techniques [22]. Although the improvements to the base material by utilizing AgNWs and CuNWs has been widely reported, review works regarding their key building applications are far from sufficient. In addition, more insights should be given into the comparison between different techniques with respect to their advantages and limitations. The objective of this paper is to provide readers a holistic review of metallic NWs from three subtopics: nanosynthesis, nanocharacterization, and specific building applications. The outline of this review paper is shown in Figure 1.

## 2. Synthesis Techniques of Metallic NWs

Metallic NWs have been synthesized by means of different shape-controlled techniques, such as template method [23], hydrothermal method [24], electro-deposition method [25], ultraviolet (UV) irradiation route [26], wet reduction method [27], chemical vapor deposition (CVD) [28], etc. Among various metallic NWs, AgNWs and CuNWs are most widely studied and exploited, since they possess similar physical and chemical properties at room temperature. Herein, the following section will give an overview of the most commonly used synthetic techniques of AgNWs and CuNWs.

### 2.1. Polyol Method

Template-assisted methods have been widely used by numerous researchers to prepare AgNWs in the past two decades, and these methods mainly fall into hard-template and soft-template methods [2]. Both techniques have advantages and disadvantages. For instance, Zhang et al. [29] pointed out that the hard template method can easily control the dimensions of AgNWs, since the template acts as a scaffold to control the diameter and length of NWs. However, the purification process is time-consuming and complicated, since the removal of the template by means of post-calcination or etching may damage the NWs [2]. Anodic alumina oxide (AAO) [30], porous silica [31], carbon nanotube [32], and block copolymers [33] have been widely used as hard templates in recent years. Details on these templates can be found in previous studies [22,34]. In contrast, the purification process of the soft template method is very convenient because the soft template can dissolve in solution. Soft template-like micelle networks can be formed by using cetyltrimethylammonium bromide (CTAB), which has been demonstrated in the work by Song et al. [35]. Nevertheless, drawbacks of some early soft-template methods [36,37] include irregular morphologies and low yield.

On this basis, Sun et al. [38] developed the polyol method to overcome the above mentioned drawbacks, which has been the most frequently used technique in terms of simplicity, cost, and large-scale synthesis of AgNWs. As a prominent soft-template technique, the main feature of the polyol process involves the formation of silver nanoparticles (AgNPs) by the reduction of silver salt using exotic PtNPs seeds at 160 °C. Generally, silver nitrate (AgNO_3_), ethylene glycol (EG), and polyvinylpyrrolidone (PVP) are used as the precursor, solvent or reducing agent, and end-capping agent, respectively. EG is suitable as the solvent owing to its high boiling point (197.3 °C) and ability to solve both PVP and AgNO_3_. When the reaction temperature is above 150 °C, EG can reduce PtCl_2_ and induce the seeding of PtNPs. Then, silver ions will be reduced to metal silver on the Pt seeds under similar conditions. The growth mechanism proposed by Sun et al. [38] is shown in Figure 2. Figure 2A indicates that the AgNW evolves from multiply twinned nanoparticles (MTPs), and therefore it has a pentagonal cross-sectional shape with a five-fold twinned structure. Chen et al. [39] pointed out that there is a great surface energy difference between the side surface (110) facets and end surface (111) facets, so that the PVP preferentially reacts with the (110) facets rather than the (111) facets as depicted in Figure 2B. In brief, PVP is an important player to not only protect the AgNWs and AgNPs from aggregating but also promote the anisotropic growth of NWs. In addition to PVP, sodium dodecyl sulfate (SDS) [40], Vitamin B_2_ [41], and CTAB [42] have been used as capping agents in some modified polyol methods. Due to its high degree of polymerization, PVP is still considered to be the best capping agent [34]. On the other hand, the platinum salt used for the prefabrication of exotic Pt seeds is very expensive, which is deemed to be a disadvantage of the early polyol method.

Since the Pt seed-assisted polyol process is not cost-effective, Sun and Xia. [43] further developed a self-seeding polyol method by simultaneously injecting 3 mL EG solution (0.1 M) of AgNO_3_ and 3 mL EG solution (0.6 M) of PVP (Mw = 55000) into preheated 5 mL EG solution at an accurate rate of 0.3 mL min^−1^ using a syringe pump. The reaction was maintained at 160 °C for 60 mins using magnetic stirring for silver ion reduction and growth of AgNWs. Shown in Figure 3A and C are the scanning electron microscopy (SEM) and transmission electron microscopy (TEM) images of uniform AgNWs, indicating that the synthesized AgNWs have diameters of 50–60 nm and lengths of 1–50 µm. The diffraction peak patterns (Figure 3B) clearly indicate the face center cubic (fcc) structure of silver. Figure 4D shows a single AgNW, in which the (110) twin plane parallel locates to its longitudinal axis.

It has also been reported that microwave-assisted polyol method is a promising route for the rapid preparation of AgNWs [44]. Tsuji et al. [45] has synthesized Ag nanorods and AgNWs by using a microwave-assisted polyol process. The authors pointed out that the average shapes, sizes, and yields of Ag nanostructures were measured as a function of the concentration of Pt, PVP, or AgNO_3_, heating time, or microwave power to determine optimum conditions for the synthesis.

A great many modified polyol methods have been proposed, but these methods can be generally divided into three stages: nucleation (reduction of silver ions to silver NPs), seeding (development of NPs to seed), and growth (growth of seed to NWs). Factors affecting the growth of AgNWs in the polyol process include reaction temperature and time [46], stirring rate [47], flow rate and molar ratio of PVP and AgNO_3_ solution [38], etc. By changing the aforementioned parameters at each synthesis stage, AgNWs with different morphologies and properties can be synthesized. In addition, it has been reported that metal ions (e.g., Fe^2+^/ Fe^3+^ [48] and Cu^2+^/ Cu^+^ [49]) and anions (e.g., Br^−^ [50] and Cl^−^ [51]) could control the diameter and growth of AgNWs. The roles played by these control agents in the polyol process can be found in a previous study [52]. Since the useful properties of metallic NWs are size-dependent, focuses have been put on obtaining AgNWs with well controlled size by fine-tuning synthesis.

### 2.2. Hydrothermal Method

Although AgNWs have shown their superior properties, but metal silver is expensive and less abundant in comparison with copper. Copper possesses similar physical and chemical properties to silver and is able to form CuNWs via different synthetic routes [53,54,55,56], among which hydrothermal method is considered as an easy-to-apply and environmentally friendly method for large-scale synthesis. In 2003, Liu et al. [24] synthesized ultralong CuNWs via the reduction of Cu^2+^-glycerol complexes by using phosphite and sodium dodecyl benzenesulfonate as the reductant and surfactant, respectively. Later on, Shi et al. [57] successfully reduced copper chloride (CuCl_2_) by using octadecylamine (ODA) as both the reducing agent and absorption agent. CuNWs with uniform diameters of 50–100 nm and an aspect ratio > 10^5^ was generated. The authors also pointed out that the diameter and aspect ratio were affected by the concentration of ODA and reaction temperature. In some cases, microwave activation was used as the heating source instead of a water bath heating for rapid synthesis of CuNWs [58] and AgNWs [59].

Recently, Kumar et al. [60] prepared ultralong CuNWs by hydrothermal method using different alkyl amines and glucose as the capping agents and reducer, respectively. Three kinds of alkyl amines, including hexadecylamine (HDA), ODA, and oleylamine (OLA), were used to investigate their roles in the growth of CuNWs. In their experiment, an aqueous solution containing CuCl_2_, glucose, and alkyl amines was stirred vigorously at room temperature overnight to obtain a blue emulsion, which was transferred into a Teflon-lined autoclave and heated to 120 °C for 24 h. Figure 4 shows the prepared CuNWs by adding HDA, ODA, and OLA. In all these cases, the lengths of the CuNWs are in the order of hundreds of microns (Figure 4A,B,D,E,G,H) and the diameters of the NWs are in the order of 20–40 nm (Figure 4C,F,I). The pentagonal ends of the NWs are shown in the insets (Figure 4B,H). A few CuNPs are associated with the CuNWs (Figure 4B,E,H), which could be easily removed by cross-flow filtration method. The authors explained that since the carbon chain length increased from HDA to ODA and OLA, their solubility in water was reduced, which would be unfavorable for the initial seeding process. Therefore, some NPs coexist with the NWs in the ODA and OLA cases.

The mechanism of the hydrothermal method is similar to the aforementioned polyol method for AgNWs synthesis. However, the AgNWs by means of polyol method usually have diameters in the order of 50–200 nm and lengths in the order of 30–80 µm [61]. To further improve the aspect ratio of AgNWs, several studies have reported using the solvothermal method to prepare ultralong AgNWs. Cwik et al. [61], for example, reported on the synthesis of ultralong AgNWs by the hydrothermal method using H_2_O_2_ and PVP as the reducer and stabilizing agent, respectively. Distribution analysis showed that the yielded AgNWs have a mean diameter of 100 nm and length of 160 µm, reaching the maximum length of 800 µm (Figure 5). In another study by Bari et al. [62], glucose was utilized as a soft reducer. The length of the AgNWs varied from 200–500 µm with an average diameter of 45–65 nm. Similarly, Zhang et al. [63] prepared AgNWs by one-pot reaction using EG as the solvent and reducing agent and high molecular weight PVP (M_w_ = 1300000) and FeCl_3_ as the capping agent. In their experiment, the reaction temperature and time were 130 °C and 8 h. Upon competition, removal of short-length AgNWs and AgNPs was completed by a filter cloth (Figure 6A). A uniform AgNW with diameter of ~55 nm and length of ~220 µm was obtained, as shown in Figure 6B–F.

A summary of the main feature, advantages, and limitations of the widely used approaches for fabrication of metallic NWs is presented in Table 1. 

## 3. Characterization of Metallic NWs

The physical and chemical properties of materials can dramatically change at the nano meter scale compared to those found in bulk materials [22]. On this basis, the characterization of metallic NWs is an important part of metallic NWs research.

### 3.1. Morphology

The morphologies of nanostructures are closely related to their properties and functions. For example, the efficiency of metallic NWs for thermal enhancement of phase change material (PCM) was found to be three times as much as the metallic NPs due to their high aspect ratio [64]. Zhang et al. [29] defined that the diameter and length of typical metallic NWs are in the range of 10–200 nm and 5–100 µm, and the aspect ratio should be higher than 10.

To investigate the morphology of metallic NWs, TEM and SEM are the most applied methods. The TEM and SEM provide useful information regarding the shape and size of metallic NWs. However it should be noted that the diameter of metallic NWs measured by SEM is usually higher than that from TEM due to the adherence of PVP on the NW’s surface [29]. High resolution TEM (HRTEM) provides the information of lattice spacing to obtain the crystal structure of metallic NWs. Other widely used methods include atomic force microscopy (AFM) and X-ray diffraction (XRD). The AFM can provide the surface topography and roughness of metallic NW networks. XRD is able to analyze the crystalline properties of metallic NWs.

### 3.2. Electrical and Thermal Conductivity

Metallic NWs have excellent electrical and thermal conductivity. Generally, the size of metallic NWs greatly affects these two properties. It has been reported that for metallic NWs with diameters below the mean free path of bulk electrons, the electrical and thermal conductivity can be significantly decreased compared to their bulk counterparts [65,66,67]. This can be attributed to the grain boundary and surface effects, which become important scattering sources. In addition, temperature is another factor affecting the Lorenz number of metallic NWs (proportionality coefficient of the ratio of thermal conductivity to electrical conductivity). For example, Cheng et al. [66] found that at 292 K, the Lorenz number of AgNW (5.2 × 10^−8^ Ω W K^−2^) was much larger than the bulk silver (2.32 × 10^−8^ Ω W K^−2^), and reduced with the decreasing temperature owing to the phonon-assisted electron scattering. He et al. [68] observed a similar phenomenon of a larger Lorenz number for AgNW, but its value was reported to decrease with the increasing temperature.

The measurements of electrical and thermal conductivity of metallic NWs are of importance to some specific applications. Until now, the electrical conductivity of a single or arrays of metallic NWs has been straightforward to measure, but it is more difficult to measure their thermal conductivity [69]. The difficulty lies in the suspension of metallic NWs and eliminating contact resistance, because heat can diffuse through any media. For a single NW, the electrical and thermal properties can be measured by powerful techniques, such as microchips (electrical based), Scanning Probe Microscopy (SPM), and optical means. It is worth mentioning that microchips can be used to measure the Seebeck coefficient to characterize the thermoelectricity of NW. For a NW array, its electrical conductivity and Seebeck coefficient can be directly measured but it is still challenging to measure the thermal conductivity. Two known methods include steady state and AC current. A thorough revision regarding the above-mentioned techniques and their merits and demerits has been presented by Rojo et al. [67].

### 3.3. Optical Property

The excellent optical properties of metallic NWs can be attributed to the plasmonic effects, which are the interactions between electromagnetic radiation (e.g., sunlight) and free electrons. The electrons will polarize and oscillate when the electromagnetic radiation hits the surfaces of metals [70]. In fact, metal nanostructures, such as Au, Ag, and Cu, possess surface plasmon resonance (SPR), which exhibits resonance light absorption in the visible region [71]. This is due to the match between the wavelength of incident light and electron oscillation and can be used as a light trapping approach for improving the solar harvesting performance of thin PV cells. The SPR of metallic NWs has been studied by using techniques such as optical spectroscopy [72] and discrete dipole approximation (DDA) [73].

It is well known that the morphology of nanostructures can affect the SPR behavior, so that it is able to examine the shape-controlled morphology of metallic NWs by means of plasmon absorption spectroscopy. Generally, Ultraviolet-visible (UV-Vis) absorption spectra is a widely used approach to measure the optical properties of metallic NWs. For example, different Ag nanostructures show different absorption peaks and intensities (Figure 7). In the case of AgNW array, the spiculate peak in the range of 343–355 nm can be attributed to the bulk silver film, while the broad peak in the range of 415–483 nm is due to the transverse mode of AgNW arrays [74]. In terms of Cu nanostructures, the absorption peak was reported to be around 570 nm by Xiong et al. [75], and CuNWs showed the highest absorption of visible light compared with other nanostructures, such as nanobelts and NPs.

## 4. Building Applications Using Metallic NWs

In this section, we present an overview of metallic NWs and their key building applications, including thermal energy storage (TES), thermal transport, electrochromic window (ECW), and photovoltaic (PV) cells. The characteristics of NWs and their performances for building applications are outlined.

### 4.1. Thermal Energy Storage

As the building sector takes up about 45% of global energy consumption, enhancements to building energy efficiency and reduced reliance on air-conditioning systems is very important to achieve a sustainable future. Phase change materials (PCMs) are substances that absorb, store, and release thermal energy isothermally, and are therefore well suited for developing energy-efficient buildings, both actively and passively. The most commonly used solid-liquid PCMs fall into organic PCMs (e.g., paraffins and fatty acids) and inorganic PCMs (e.g., salt hydrates). A comprehensive review regarding the building application of PCMs has been presented by Zhou et al. [76]. However, the applications of PCMs are usually limited by their inherent low thermal conductivity, which could be improved by dispersing thermally-conductive metallic NWs as the thermal conductivity promoter.

CuNWs, with their high thermal conductivities, have been used to develop nano-enhanced PCMs for building applications. Shah et al. [14] prepared CuNWs by disproportionation of a Cu^+^ precursor in ODA. After purification, reddish nanostructures were obtained, and their structures were verified by TEM and XRD, as shown in Figure 8. The prepared CuNWs were incorporated into a hydrated CaCl_2_∙6H_2_O salt-based PCM and the corresponding thermal conductivities at different concentrations of CuNWs were measured. Their results showed that just 0.02 wt.% of CuNWs increased thermal conductivities by >20%, although it should be noted that the effect of diminishing returns is also observed past 0.08 wt.% of CuNWs (Figure 9).

Zeng et al. [64] synthesized ultralong CuNWs and dispersed them into tetradecanol (TD) as the base PCM. The SEM image (Figure 10) shows that the TD-based PCM was absorbed within the sponge-like structure of CuNWs. The Differential Scanning Calorimetry (DSC) results showed that the endothermic peak of the composite PCM was sharper than that of TD due to the enhanced thermal conductivity of CuNWs (Figure 11A). In addition, the enthalpy changes (ΔH) of the composite PCMs were calculated and found to be linear (Figure 11B). Thermogravimetry (TG) tests showed that the pure TD underwent a single step weight loss, corresponding to the evaporation of TD during the heating process. The composite PCMs lost weight slower than that of TD due to the absorption of TD in the voids of CuNWs (Figure 11C). The thermal conductivity measurements of the composite PCMs showed that a relatively linear relationship between the thermal conductivity and loading of CuNWs existed. The composite PCM containing 11.9 vol% of CuNWs showed a thermal conductivity of 2.86 W/m K, which was nearly 9 times higher than that of TD only (0.32 W m^−1^ K^−1^) (Figure 11D).

Similarly, Zhu et al. [77] fabricated a series of palmitic acid (PA)/polyaniline(PANI) form-stable PCMs with and without adding CuNWs. The randomly dispersed CuNWs improved the thermal conductivity of the pure PCMs from 0.377 W m^−1^ K^−1^ to 0.455 W m^−1^ K^−1^ with 11.2 wt.% of CuNWs, as shown in Figure 12. In addition, the melting heat of the composite PCMs could still reach 149 J g^−1^ in the presence of 11.2 wt.% of CuNWs.

In terms of using AgNWs as a nanopromoter, Zeng et al. [78] successfully fabricated a TD/AgNW composite PCM, which showed a remarkable thermal conductivity of 1.46 W m^−1^ K^−1^ at a volume fraction of AgNWs of 11.8%. The authors pointed out that the strong thermal enhancement is owing to the high aspect ratio of AgNWs, few thermal conduct interfaces, and high interfacial thermal conductivity.

Recently, Deng et al. [79] fabricated a series of novel shape-stabilized composite PCMs (ss-CPCMs) by impregnating polyethylene glycol (PG)-wrapped AgNWs into the pores of expended vermiculite (EVM). In these composites, polyethylene glycol (PEG), AgNWs, and EVM were used as the base PCM, thermal conductivity promoter, and shape stabilizer, respectively. Polyol method was used to prepare the AgNWs showing a length of 5–20 µm and a diameter of 50–100 nm. AgNWs wrapped by PEG were well-dispersed and enwrapped inside the pores and surfaces of EVM (Figure 13). It was indicated that the maximum encapsulation capacity of PEG in all cases with good shape stability was 66.1 wt.%. The EVM also reduced the supercooling degree of PEG by about 7 °C, since it serves as a nucleation to promote the crystallization of PEG (Figure 14A). The thermal conductivity of AgNW (19.3 wt.%)-enhanced composite PCM reached 0.68 W/m K, which was almost 11.3 times higher than that of the pure PEG (Figure 14B). A summary of AgNWs and CuNWs for thermal conductivity enhancement of PCM is presented in Table 2.

### 4.2. Thermal Transport

One of the most important components in a building system is the air-conditioning, which relies on the heat transfer between coolant and ambient air. However, most coolants, such as water, possess low thermal conductivity, leading to an increase in the system cost, since large surface area (e.g., mounting fins) is needed to ensure a high heat transfer efficiency. A similar situation also occurs in the harvesting of solar thermal energy. On this basis, there is a need to enhance the thermal transport of the base heat transfer fluid.

Carbajal–Valdez et al. [80] prepared AgNWs via polyol method. SEM imaging (Figure 15A) clearly shows that the uniform AgNWs had a length of up to 10 µm with a mean diameter of 96.04 nm. The AgNWs with different volume fractions were then dispersed into distilled water to form a nanofluid. It was pointed out that the AgNWs showed an effective thermal transport property at low loadings. For example, by adding 1.74 × 10^−4^ vol.% of AgNWs, an enhancement of 20.8% in thermal conductivity was achieved (Figure 15B).

Along the same line, Zhang et al. [81] prepared two kinds of AgNWs by using PVP with different molecular weights (40,000 and 130,000). It was shown that the morphology of the produced AgNWs is highly affected by the molecular weights. AgNWs produced by PVP_MW=40000_ showed a mean aspect ratio of 25 and homogeneous length of 1 µm. In the case of PVP_MW=130000_, most of the produced Ag nanostructures were NPs. The authors investigated the thermal enhancement performance of two kinds of AgNWs and found that 0.46 vol.% of AgNWs (PVP_MW_ = 40,000) could increase the thermal conductivity of base fluid (EG) by 13.42% compared to 4.76% by AgNWs (PVP_MW_ = 130000). The better performance can be attributed the higher aspect ratio of AgNWs.

It has been reported that the dispersion of nanopromoters in the base fluid will degrade the fluidity due to the increased viscosity. Hence, thermal and hydrodynamic performance of an AgNW-based nanofluid were investigated by Şimşek et al. [82]. The authors used polyol method to prepare the AgNWs, which showed a diameter of 50–100 nm and length of 5–18 µm. The convective heat transfer coefficient and pressure drop of a water-based nanofluid flowing through microchannel heat sinks were then experimentally measured. A maximum enhancement of 56% in the heat transfer coefficient was achieved at 180 µL min^−1^ in the 70 µm × 50 µm microchannel without increase in the pumping power. Table 3 summarizes the application of AgNWs for improving the thermal transport of the base fluid.

### 4.3. Electrochromic Window

Electrochromic window (ECW) technologies can provide a flexible control mode in terms of solar heat gains, natural light, and glare in indoor environments. [83]. This flexibility can be achieved by adjusting the transmittance of glazing (e.g., from tint to bleach state) through applied voltage. The ECW is beneficial for building energy efficiency and IEQ [84]. Transparent conductive electrodes (TCEs) are important components in ECW and they are usually made of indium tin oxide (ITO) and fluorine-doped tin oxide (FTO). However, they were shown to be highly brittle, and thus not suitable for flexible TCEs [85]. On the other hand, ITO and FTO are expensive and the indium resources are becoming less abundant due to the increased consumption [86]. On this basis, AgNW can be applied to fabricate TCEs due to their high conductivity, good transparency, and excellent flexibility.

Lin et al. [87], for example, developed a nonheated roll-to-roll (R2R) technique to produce flexible, ultra-large, and transparent AgNW network electrode film. The main feature of this technique is that the precursor solution was rapidly stretched into NWs by UV irradiation and by adding PVP. The reduction of Ag ions only took 3 h without heating. Figure 16 shows that amorphous carbon skeleton was attached on the surface of AgNW, which is due to the degradation of polymer and PVP. The authors pointed out that the skeleton can offer sufficient mechanical strength without affecting the electrical conductivity of AgNWs. The AgNW/polyethylene terephthalate (PET) film showed a sheet resistance of ~15 Ω sq^−1^ and transmittance of 95%. In comparison to ITO ECW, the AgNW ECW showed a shorter switching time of coloration (4.3 vs. 5.4 s), a higher electron-transfer rate, and a higher coloration efficiency (120 vs. 80 cm^2^ C^−1^), as shown in Figure 17.

It has been reported that nanosilver-based TCE is subject to oxidation of Ag under the long switching time of ECW application [88]. To overcome this downside, Mallikarjuna and Kim [89] coated a reduced graphene oxide (RGO) layer on the surface of the AgNW network. The hybrid RGO/AgNW/metal grid/PET (RAM) TCE sheet shows a resistance of 0.714 Ω sq^−1^, as well as a transmittance of 90.9% at wavelength of 550 nm. The hybrid TCE was coated with a WO_3_-based electrochromic device (ECD), as shown in Figure 18. Compared with other ECDs, the RAM-ECD showed the best switching performance with coloration efficiency of 33.4 cm^2^ C^−1^ and switching response of 12 s for coloration and 10 s for bleaching.

Similarly, Kim et al. [90] successfully fabricated a graphite/graphene/AgNW-based TCE. The TCE consisted of few-layered graphite, single layered graphene sheets, and AgNWs forming a fused junction under optimal heat treatment at 80 °C. The sheet resistance was measured to be 24–28 Ω sq^−1^ with a transmittance of 58–67%. The authors explained that the lower optical transmission was due to the scattering of incident light by the graphene layer. An ECD using the hybrid TCE showed a maximum optical contrast of 15% at the wavelength of 630 nm. Cyclic voltammetry (CV) test of the ECD showed an excellent stability without performance degradation after 500 cycles (Figure 19).

Yuksel et al. [91] prepared a AgNW/polymer (P1C)-based ECD as a blue transmissive switching material. The P1C showed a very fast switching response of 0.86 s for tinting and 0.57 s for bleaching. An AgNWs with a high aspect ratio was used as the transparent contact for the P1C layer. SEM imaging shows that the AgNW network was uniformly covered with the P1C, forming a nanocomposite TCE (Figure 20A). A good optical contrast of 24% was demonstrated for the AgNW/P1C-based ECD compared to the ITO/P1C ECD of 30%, as shown in Figure 20B.

Zhou et al. [92] first reported on the application of AgNW/WO_3_-based ECW in modulation of near-infrared (NIR) light. AgNWs with average diameter of 100 nm and length of 70 µm were fabricated by using the polyol method. The AgNWs were spun on polyvinyl alcohol (PVA) coating to form AgNW TEC. Then, electrochromic WO_3_ films were coated onto the AgNW TEC to form a three-electrode system (PVA, AgNW, and WO_3_), as shown in Figure 21A. XRD pattern (Figure 21B) indicated the amorphous state of WO_3_ film. The WO_3_/AgNW electrochromic films revealed an obvious advantage on NIR modulation over conventional WO_3_/ITO electrodes, showing excellent cycling stability and durability (Figure 22). In addition, at the NIR region of 1100 nm, the WO_3_/AgNW film showed high coloration efficiency of 86.9 cm2 C^−1^. A summary of AgNWs for fabricating TCE used in ECDs is presented in Table 4.

### 4.4. Photovoltaic Cell

In recent years, photovoltaic (PV) cells as a novel sustainable energy harvesting technology have been widely implemented to provide a viable alternative to fossil fuels [93]. Among a variety of PV cell technologies, Building Integrated Photovoltaics (BIPV) is seen as one of the most promising contributors to energy-efficient buildings [94]. BIPV can replace some building materials, such as windows or roofs, while providing aesthetic appeal for architects. It is well known that TCE is a vital component in PV cells, which directly affects the photo-conversion efficiency [95]. However, the most critical barriers for the application of BIPV are related to long term payback period and high initial costs [96], which can be partially attributed to the expensive ITO-based TCEs. On this basis, metallic NWs can be used to fabricate high performance TCEs. It is worth mentioning that there are different techniques for coating the metallic NWs on TCE film, among which solution-based methods are most widely used, such as spray coating, dip coating, doctor blade coating, etc. A summarization of these coating methods has been summarized by Basarir et al. [2].

To overcome the drawbacks of ITO, such as high cost, scarcity, and brittleness, Kim et al. [97] have fabricated hybrid PV devices by a combination of poly(3-hexylthiophene) (P3HT) and [6,6]-phenyl C61-butyric acid methyl ester (PCBM) with ratio of 1:0.7. 20 wt.% of AgNWs, which was spin coated into the active to improve the PV performance, as shown in Figure 23. The AgNW-embedded PV device showed a power conversion efficiency (PCE) of 3.91%, high short-circuit current density (J_SC_) of 9.32 mA cm^−2^, fill factor (FF) of 63.6%, and open-circuit voltage (V_OC_) of 0.66 V.

Poly(3,4-ethylenedioxythiophene) (PEDOT):poly(styrene sulfonate) (PSS) thin films embedded with AgNWs show low sheet resistance, which can be used to enhance film conductivity without much loss of transparency. For example, Lu et al. [98] fabricated a direct inkjet-printed transparent AgNW network as the top contact for organic photovoltaic (OPV). The OPV device was formulated by printing AgNW ink on the surface of the PEDOT:PSS:MoO_3_ layer. With the increase of printing times from 3 to 9, the sheet resistance of the film decreased dramatically to 26.4 Ω sq^−1^, while the transmittance of the film decreased only slightly from 95% to 83%. The best PCE of 2.71%, J_CS_ of 8.44 mA cm^−2^, FF of 54%, and V_OC_ of 0.6 V were achieved for the P3HT:PCBM-based OPV having 7-times printed AgNW. It was pointed out that the performance degradation of the device with 9-times printed AgNW was due to the undesirable solvent effect on the top of the PEDOT:PSS:MoO_3_ layer.

Similarly, Maisch et al. [99] printed AgNWs with an average length of 10 µm using industrial printheads having nozzle diameters in the same size range as the AgNW length. The inkjet-printed AgNW networks showed uniform distribution and a good balance between conductivity and optical transparency. The TCE exhibited a sheet resistance of 18 Ω sq^−1^ and a transmittance of 93% for four-times inkjet printing. The AgNW/PEDOT:PSS based device demonstrated the highest PCE of 4.3% so far for fully inkjet-printed OPV cells.

Recently, Park et al. [100] reduced the maximum processing temperature of AgNW fusing to only 120 °C by using water-based PEDOT:PSS. Spary-coated AgNW electrodes with 40 nm PEDOT:PSS were prepared on glass and PET substrates by an annealing process at 120 °C for 15 min. The TCE had a sheet resistance of 30 Ω sq^−1^ and a transmittance of 86%. OPV cells with the AgNW/ PEDOT:PSS transparent electrodes (Figure 24) showed a best PCE of 7.15%, J_CS_ of 12.34 mA cm^−2^, FF of 60.67%, and V_OC_ of 0.955 V.

Form an application point of view, it is needed to develop PV systems that can be applied on opaque substrates. An interesting example of the fabrication of OPV onto steel was recently reported by Ding et al. [101]. Shown in Figure 25 is the proposed OPV device. To enable OPVs on steel, an epoxy SU-8 is spin-coated onto a low carbon steel (DC01) substrate to ensure no electrical shorts. The AgNWs with PEDOT were spray coated onto the P3HT:PCBM layer. The transparent electrode layer demonstrated a sheet resistance of 30 Ω sq^−1^ and 90% transmittance. The low carbon-steel-based OPV device showed a PCE of 2.3%, J_CS_ of 8.2 mA cm^−2^, FF of 53%, and V_OC_ of 0.53 V.

One challenge facing the application of an AgNW network to PV device is the reduced charger carrier transport due to the empty lateral space between the AgNWs and the top surface of the device. Chung et al. [102] developed a gentle, low temperature, and chemically benign technique to fabricate AgNW/ITO-NP TCE, which is able to reduce the possibility of damaging the CdS and CuInSe_2_ layers during the deposition process. The layer of the TCE is shown in Figure 24. The AgNW/ITO-NP films were spin coated onto the device. It can be seen that the ITO-NP effectively surrounded the AgNWs, improving the electrical contact between the AgNW network and the active layer. The TCE exhibited a sheet resistance of 30 Ω sq^−1^ and a high transparency of 90%. The PV film with AgNW/ITO-NP top contacts showed a high PCE of 10.3% (Figure 26), J_SC_ of 30.1 mA cm^−2^, FF of 69.3%, and V_OC_ of 0.494 V. It was pointed out that the high PCE was due to light scattering in the AgNW/ITO-NP layer and higher fraction of carriers being successfully collected and separated into the photocurrent.

With the same purpose, Singh et al. [103] demonstrated percolation AgNWs networks covered with scalable and solution-processed ZnO film as TCE for CdS/CuInSe_2_ PV cell. The fabricated TCE showed a sheet resistance of 11 Ω sq^−1^ and transmittance of 90%. The maximum PCE of 13.4%, J_SC_ of 33.8 mA cm^−2^, FF of 62%, and V_OC_ of 0.64 V were achieved for low cost PV cells.

Another factor affecting the performance of AgNW-based TCE is the synthesis parameters of AgNWs. Teymouri et al. [104], for example, investigated the performance of TCEs prepared using two kinds of AgNWs. The two AgNWs were fabricated by modified polyol (MP) approach and Sigma polyol (SP) approach, respectively. The MP-AgNW-based TCE exhibited a very low sheet resistance of 11.3 Ω sq^−1^, whereas the SG-AgNW-based TCE showed a high sheet resistance of 99.3 Ω sq^−1^. The prepared PV cell using MP-AgNW-based TCE showed a PCE of 3.18%, which was 28% higher than the one using SP-AgNW-based TCE. A similar study has also been reported by Wei et al. [105].

CuNWs can be also used to fabricate TCE with the advantage of being cheaper than the AgNWs. However, the oxidation layer on the surface of CuNWs will degrade the electrical conductivity. To solve this problem, Won et al. [106] applied lactic acid to etch the organic capping molecule and the surface oxide/hydroxide from the CuNWs. This treatment enabled direct contact between CuNWs, improving the transport of electrons. The acid-treated CuNWs TCE film showed a sheet resistance of 19.8 Ω sq^−1^ and transmittance of 88.7% at 550 nm. The Al-doped ZnO (AZO) dramatically improved the thermal stability and oxidation resistance of CuNWs. In addition, the AZO/CuNW/AZO TCE film showed a high PCE of 7.1%, J_SC_ of 33.7 mA cm^−2^, FF of 36.0%, and V_OC_ of 0.58 V. Table 5 summarized the application of AgNWs and CuNWs for fabricating TCE used in PV cells.

## 5. Conclusions

Metallic NWs are multifunctional materials that can be positively utilized to improve building performance by modifying the vital properties of base materials. In this review, the most widely applied wet-chemistry synthesis techniques of AgNWs and CuNWs have been introduced, highlighting the factors affecting the products’ properties. The ability to control the morphology and distribution of metallic NWs during synthesis remains an important issue for the future. In addition, characterization of metallic NWs, including morphology, electrical and thermal conductivity, and optical properties, have been overviewed. AgNWs are found to be frequently used for thermal enhancement of PCM and nanofluids and fabrication of TCEs in ECW and PV cells. Applications of AgNW-enhanced base materials are very useful in facilitating energy-efficient building design, particularly for PCM and ECW. Although numerous synthesis techniques have been proposed, further efforts are still needed to prepare thinner, longer, and uniform AgNWs and CuNWs, since high aspect ratio is beneficial for both thermal and electrical transport. On one hand, ease of oxidization of AgNWs and CuNWs is a problem facing their application in the TCE industry, which merits further efforts to prepare oxidation-resistant NWs. On the other hand, agglomeration and sedimentation of AgNWs and CuNWs may occur in the nanofluid or PCM after a great number of thermal cycles, which requires the use of surfactant to achieve homogeneous dispersion. Furthermore, the improved building performance by using these advanced metallic NWs and their products are still lacking in the published literature. This means more efforts are needed to examine the practical performance of metallic NWs for the construction industry. Finally, we envisage that the field of metallic NWs in the building industry will be gradually transformed from a budding stage to a blossoming stage, and more fundamental research is, therefore, required to exploit other superior metallic NWs.

## Figures and Tables

**Figure 1 materials-12-01731-f001:**
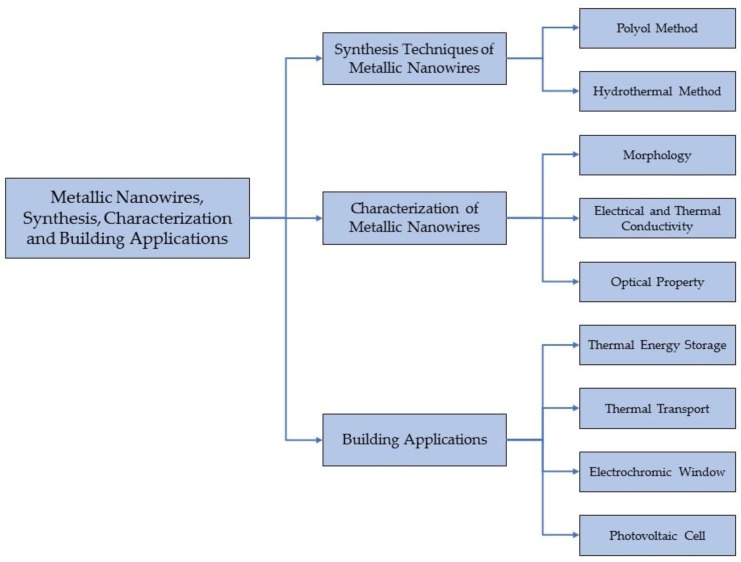
Overview of the review paper.

**Figure 2 materials-12-01731-f002:**
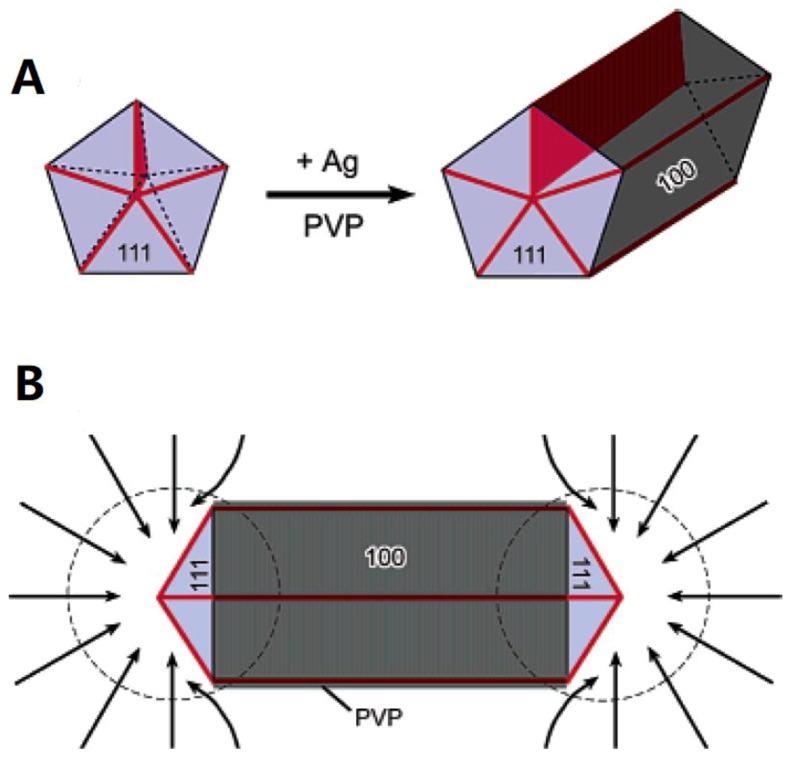
Growth mechanism of silver nanowires (AgNWs) with pentagonal cross sections: (**A**) evolution of a nanorod from a multiply twinned nanoparticles (MTPs) of silver and (**B**) diffusion of silver atoms toward the two ends of a nanorod, with the side surfaces completely passivated by polyvinylpyrrolidone (PVP) [38].

**Figure 3 materials-12-01731-f003:**
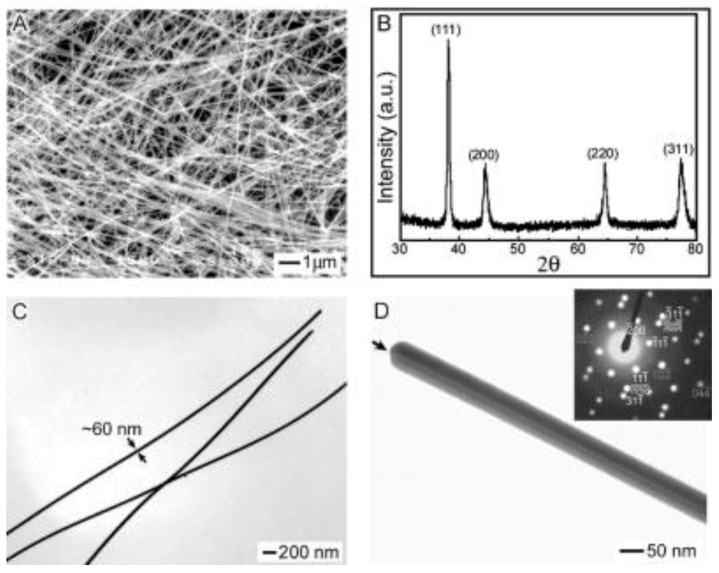
(**A**) Scanning electron microscopy (SEM) and (**C**) transmission electron microscopy (TEM) image of homogeneous AgNWs prepared using polyol method, (**B**) X-ray diffraction (XRD) pattern indicating face centered cubic (fcc) structure, and (**D**) TEM image showing the twin plane of single AgNWs and microdiffraction pattern (inset) [43].

**Figure 4 materials-12-01731-f004:**
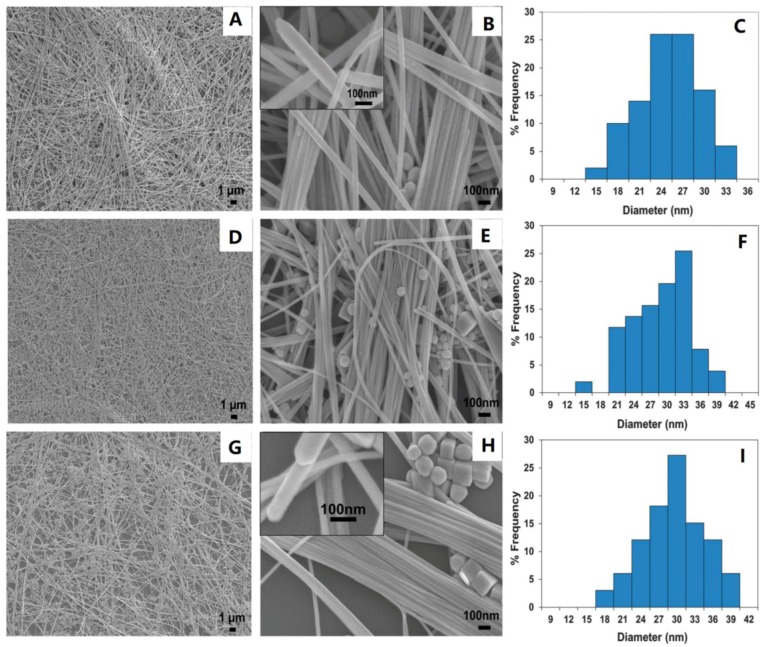
Field-emission scanning electron microscope (FESEM) images of copper nanowires (CuNWs) prepared using (**A**,**B**) hexadecylamine (HDA), (**D**,**E**) octadecylamine (ODA), and (**G**,**H**) oleylamine (OLA). The diameter distributions of the corresponding CuNWs prepared by using HDA, ODA, and OLA are shown in (**C**), (**F**), and (**I**), respectively. The insets in (B) and (H) show side views of the nanowires with pentagonal ends [60].

**Figure 5 materials-12-01731-f005:**
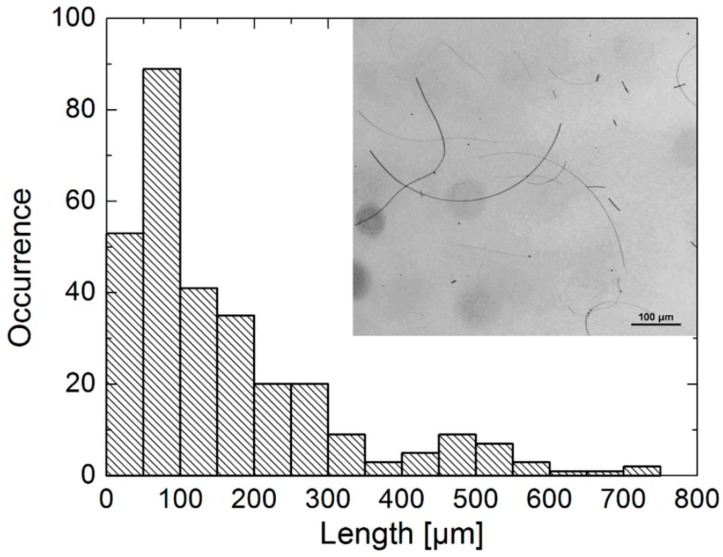
Length Distribution of AgNWs after the synthesis. An example of the reflection bright-field image is inset [61].

**Figure 6 materials-12-01731-f006:**
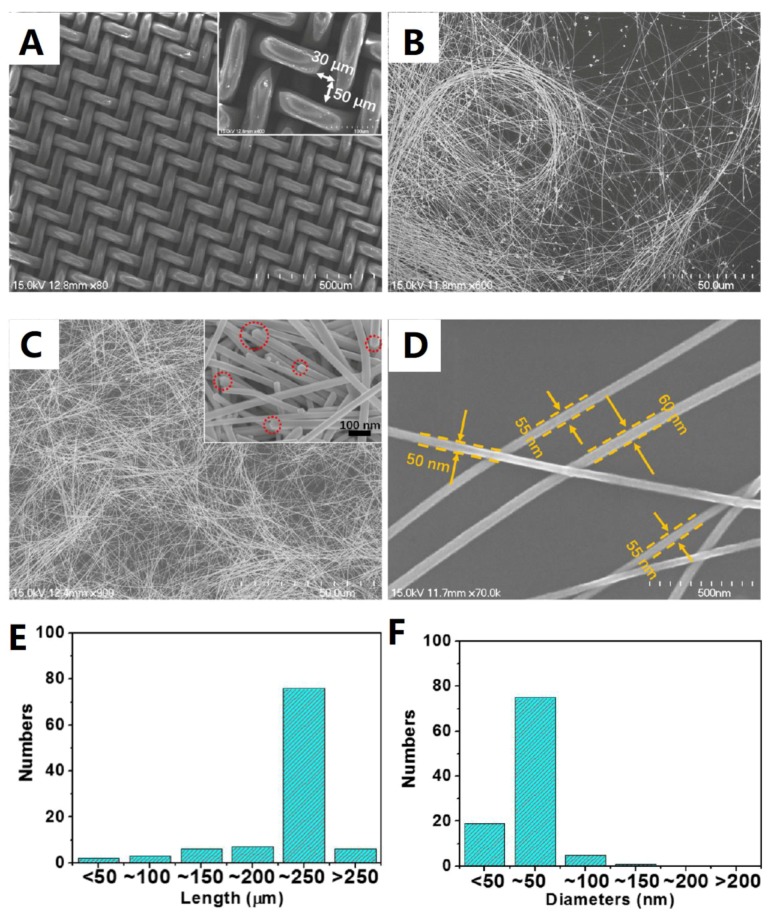
(**A**) Filter cloth used to remove short-length nanowires (NWs) and nanoparticles (NPs), SEM images of AgNWs before (**B**) and after (**C**) filtration with inset showing the penta-twinned structure (**D**) SEM image of AgNWs, the distribution of (**E**) length, and (**F**) diameters [63].

**Figure 7 materials-12-01731-f007:**
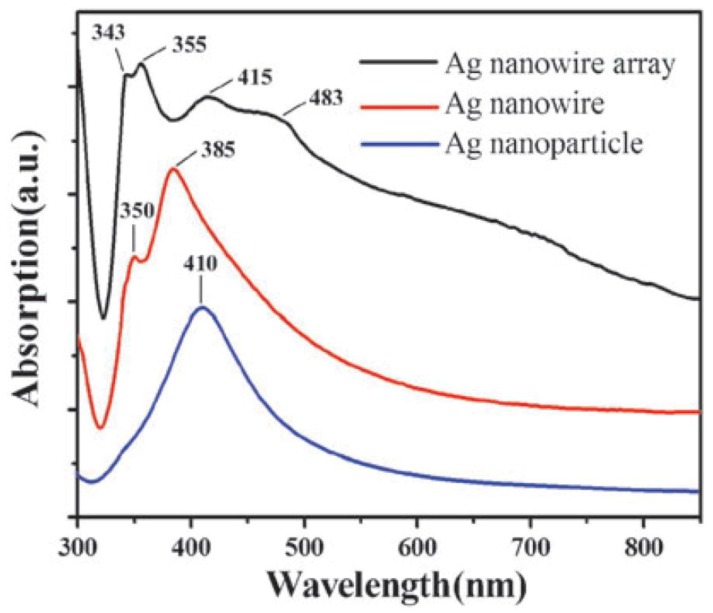
Ultraviolet-visible (UV-vis) absorption of different silver nanostructures [74].

**Figure 8 materials-12-01731-f008:**
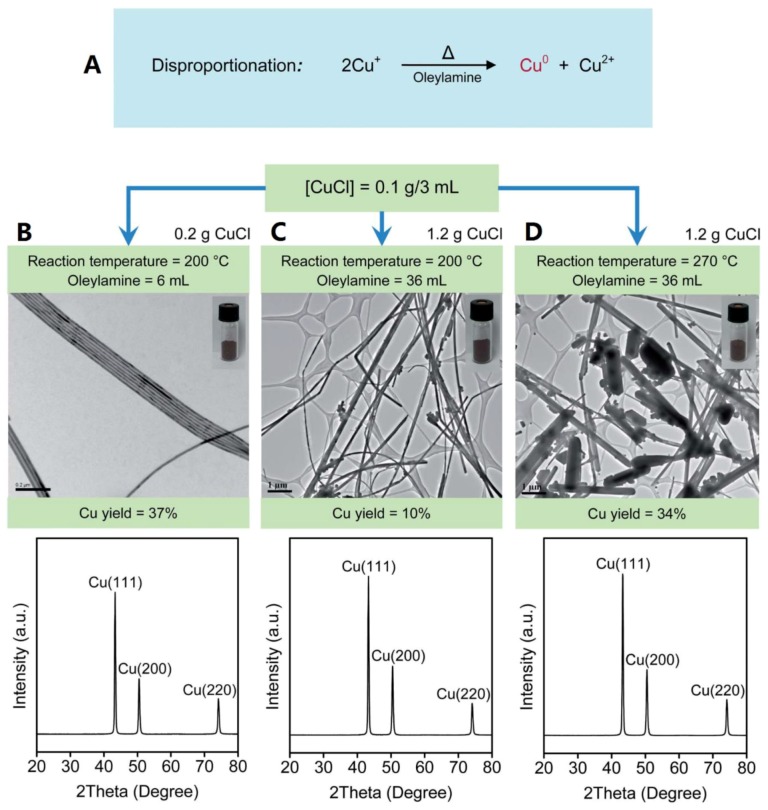
(**A**) Reaction for generating CuNWs. (**B**–**D**) Yield, morphology, and XRD pattern of CuNWs prepared at different conditions. Insets show the reddish CuNWs products after purification [14].

**Figure 9 materials-12-01731-f009:**
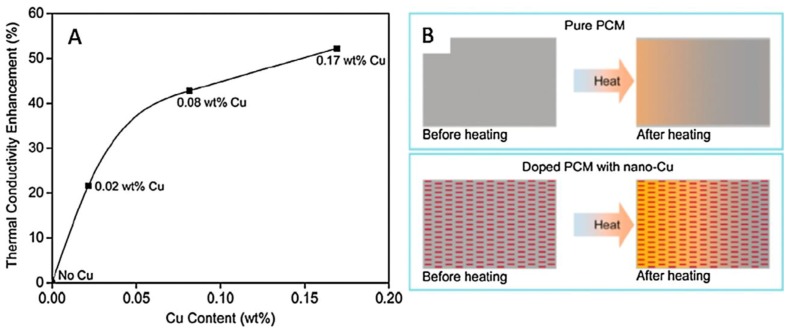
(**A**) Thermal conductivity enhancement of phase change material (PCM) (**B**). Schematic diagrams of heat transfer and dissipation in the pure PCM versus CuNW-doped PCM upon heating [14].

**Figure 10 materials-12-01731-f010:**
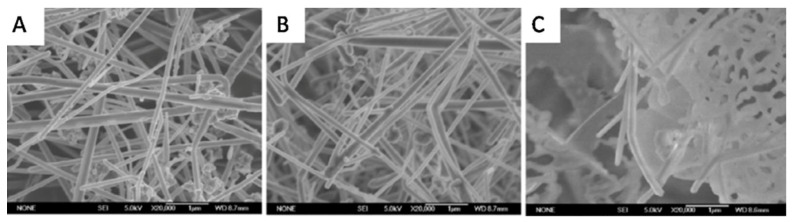
SEM images of CuNW (**A**) composite PCM with 58.9 w.t% (**B**) and 1.32 wt.% (**C**) of CuNWs [64].

**Figure 11 materials-12-01731-f011:**
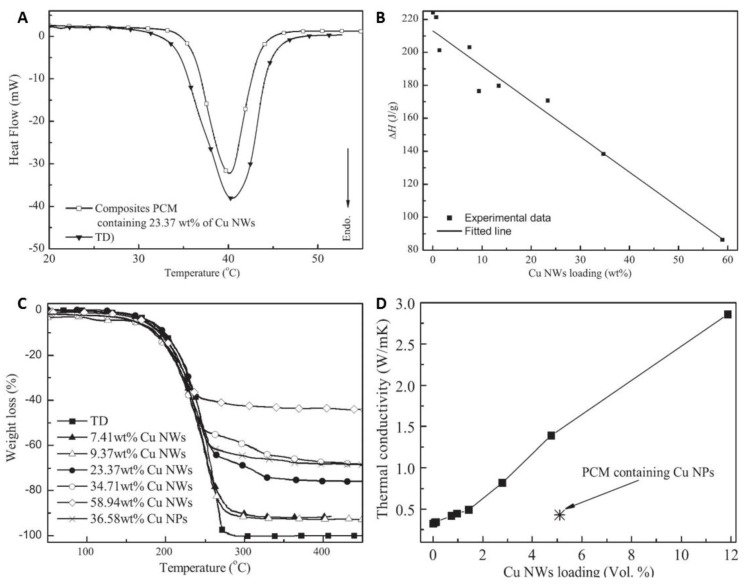
(**A**) differential scanning calorimetry (DSC) curves, (**B**) phase change enthalpy (ΔH) with different loadings of CuNWs, and (**C**) thermogravimetry (TG) curves of tetradecanol (TD) and the composite PCMs. (**D**) Thermal conductivity of the composite PCMs [64].

**Figure 12 materials-12-01731-f012:**
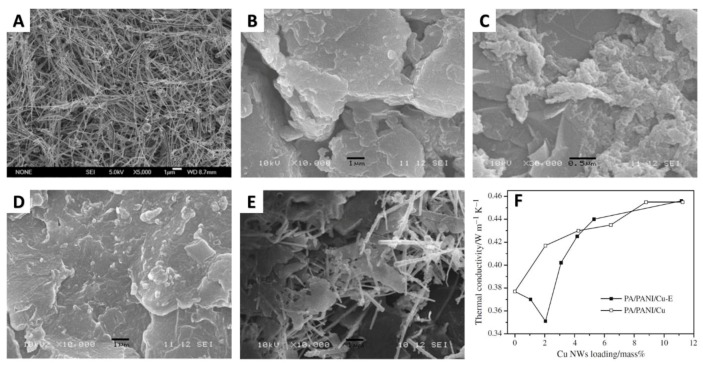
SEM images of (**A**) CuNWs, (**B**) palmitic acid (PA)/polyaniline (PANI)-2, and (**C**) PA/PANI-2 was washed with ethanol three times. (**D**) PA/PANI/Cu-E-6 and (**E**) PA/PANI/Cu-5 (**F**) thermal conductivity of the form-stable PCMs doped with CuNWs [77].

**Figure 13 materials-12-01731-f013:**
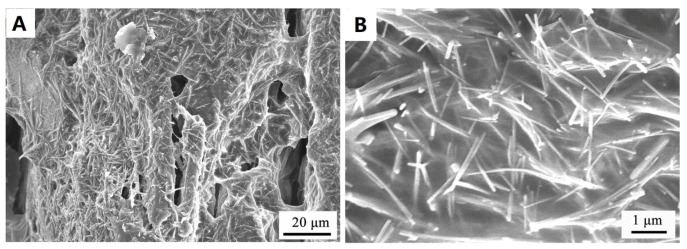
SEM images of polyethylene glycol (PEG)–Ag/expended vermiculite (EVM) shape-stabilized composite PCMs (ss-CPCM) containing 19.3 wt.% of AgNWs with scale bar of (**A**) 20 µm and (**B**) 1 µm [79].

**Figure 14 materials-12-01731-f014:**
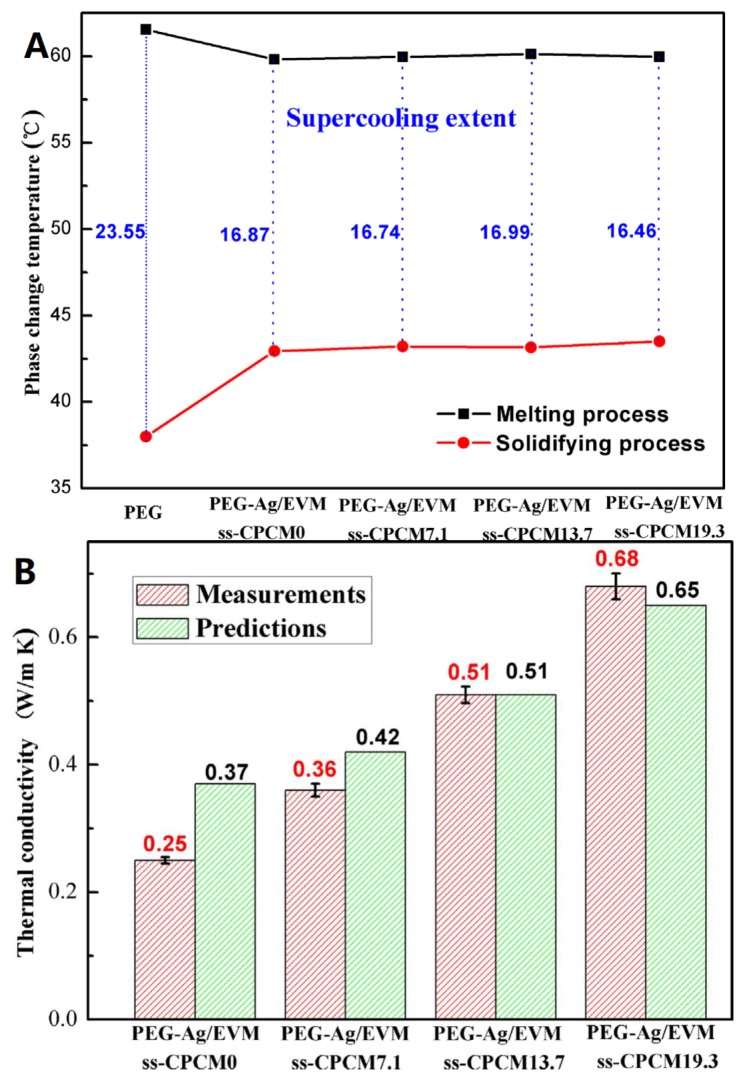
(**A**) Supercooling degrees of PEG and PEG-Ag/EVM ss-CPCMs with different concentration of AgNWs and (**B**) measured and predicted thermal conductivities of PEG-Ag/EVM ss-CPCMs at 20 °C [79].

**Figure 15 materials-12-01731-f015:**
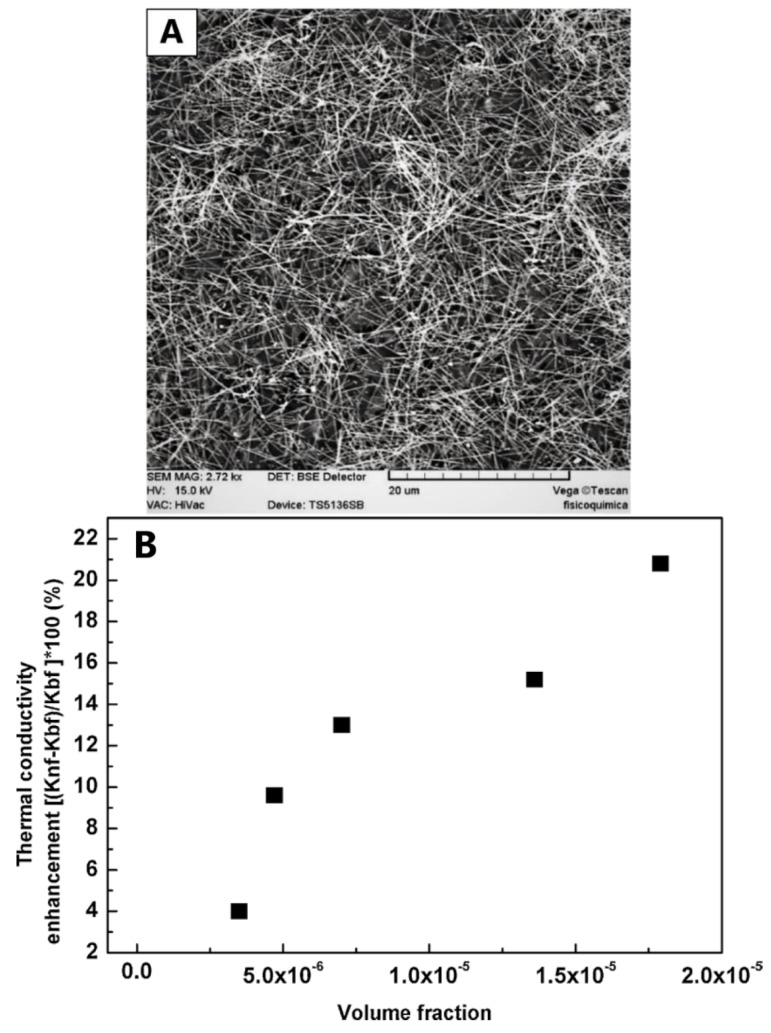
(**A**) SEM image of AgNWs (magnification 2750) and (**B**) thermal conductivity enhancement of nanofluids with different volume fractions of AgNWs [80].

**Figure 16 materials-12-01731-f016:**
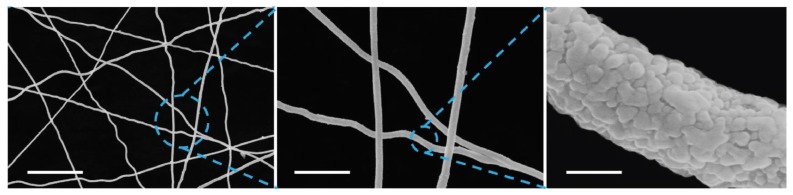
SEM image of AgNW network (scale bar 10 and 2 µm and 300 nm) [87].

**Figure 17 materials-12-01731-f017:**
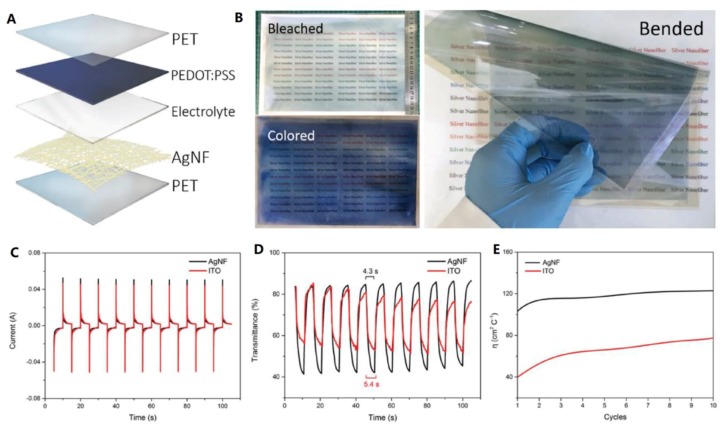
(**A**) structural schematic of AgNW-based electrochromic window (ECW), (**B**) flexible A4-sized ECW, (**C**) current response, (**D**) transmittance response, and (**E**) coloration efficiency during the first 10 switching cycles [87].

**Figure 18 materials-12-01731-f018:**
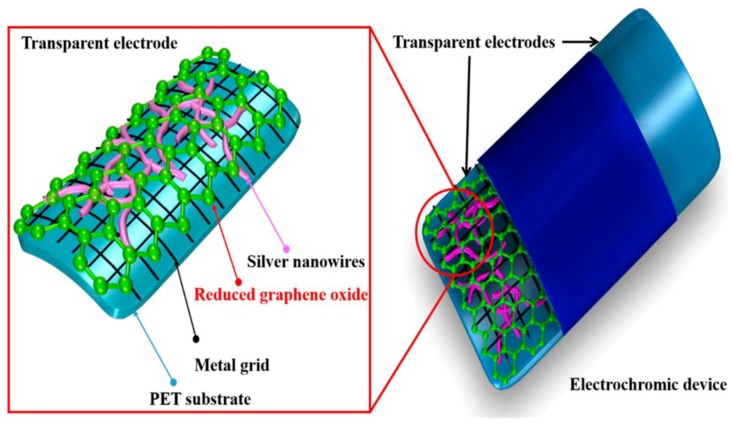
Schematic diagram of RGO/AgNWs/metal grid/polyethylene terephthalate (PET) electrode-based WO_3_ electrochromic device (ECD) [89].

**Figure 19 materials-12-01731-f019:**
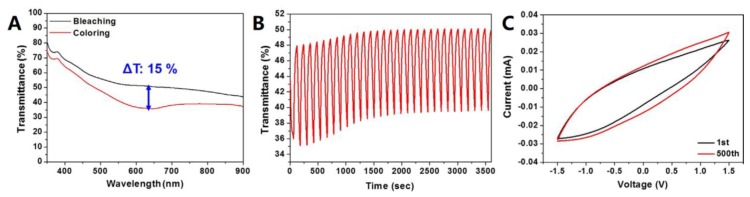
Characterization of poly(3,4-ethylenedioxythiophene) (PEDOT):poly(styrene sulfonate) (PSS)-TiO_2_ ECD: (**A**) total transmittance change of ECD before and after coloration, (**B**) cyclic transmittance change of ECD at 630 nm, and (**C**) cyclic voltammograms [90].

**Figure 20 materials-12-01731-f020:**
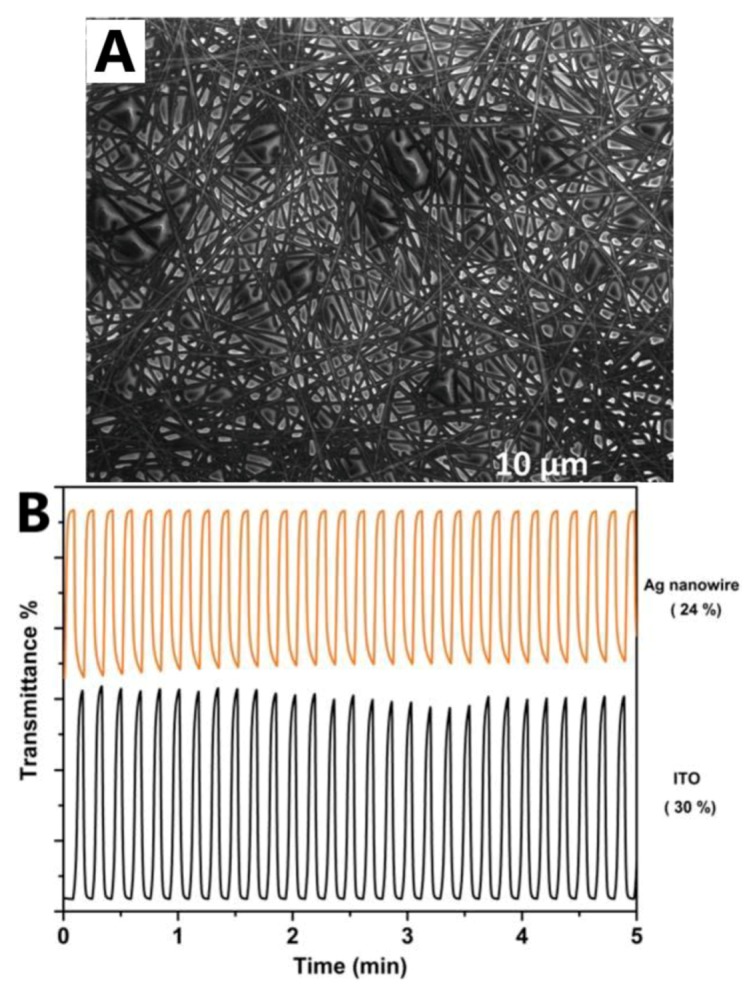
(**A**) SEM image of Ag/NW/P1C nanocomposite electrodes and (**B**) cyclic voltammetry (CV) study of P1C on AgNW-network-deposited PET substrates [91].

**Figure 21 materials-12-01731-f021:**
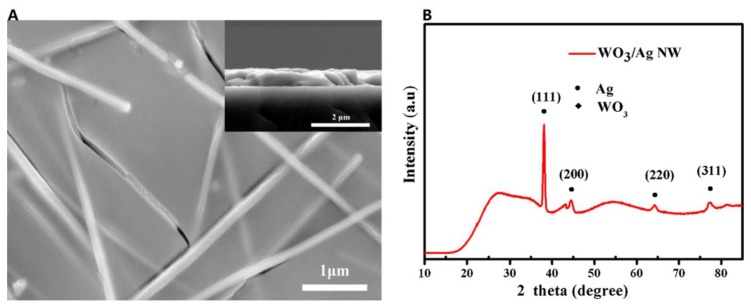
(**A**) SEM image of the AgNW electrode coated with WO_3_ film and the cross-section of the WO_3_/AgNW (inset) and (**B**) XRD pattern of the WO_3_/AgNW film [92].

**Figure 22 materials-12-01731-f022:**
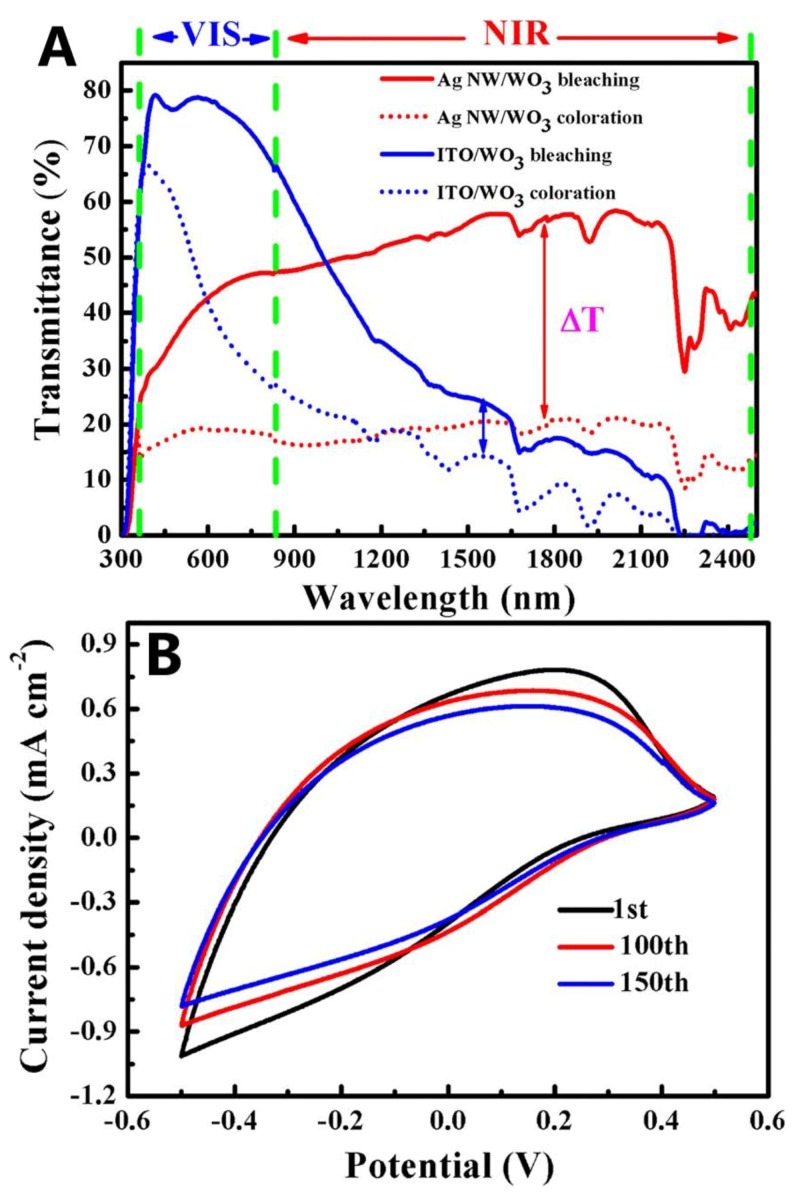
(**A**) Transmittance spectra of the ECDs using conventional ITO substrate and AgNW substrate in their colored and bleached states and (**B**) cyclic voltammograms for AgNW-based ECD with scan speed of 50 mV s^−1^ at the 1st, 100th, and 150th cycle [92].

**Figure 23 materials-12-01731-f023:**
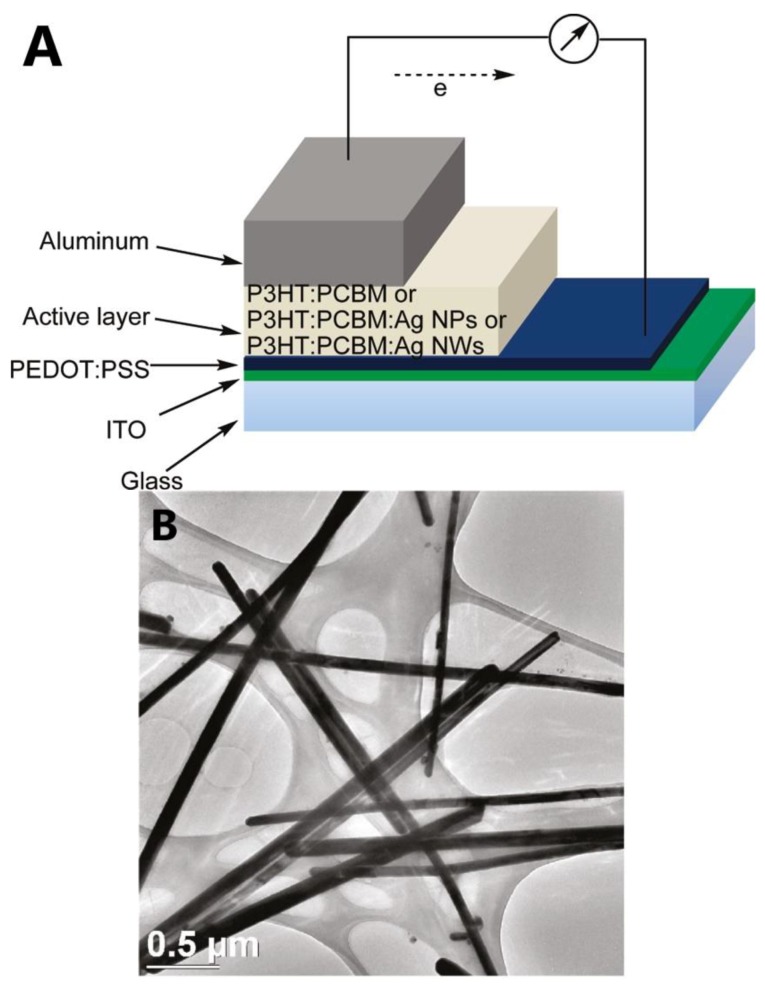
(**A**) Layout of PV device and (**B**) TEM image of poly(3-hexylthiophene) (P3HT)-[6,6] phenyl C61-butyric acid methyl ester (PCBM):AgNWs structure [97].

**Figure 24 materials-12-01731-f024:**
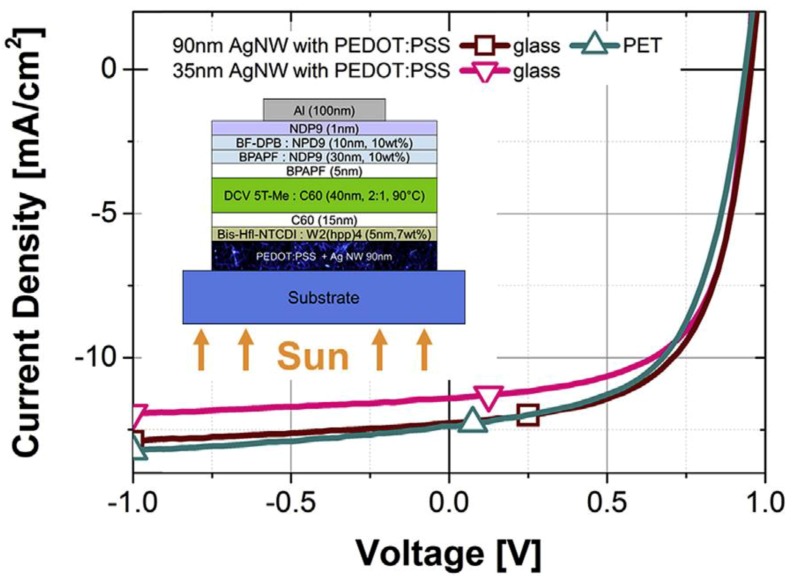
Current density-voltage (J-V) curve of organic photovoltaic (OPV) device using AgNW/PEDOT:PSS transparent electrodes, with inset showing the device structure [100].

**Figure 25 materials-12-01731-f025:**
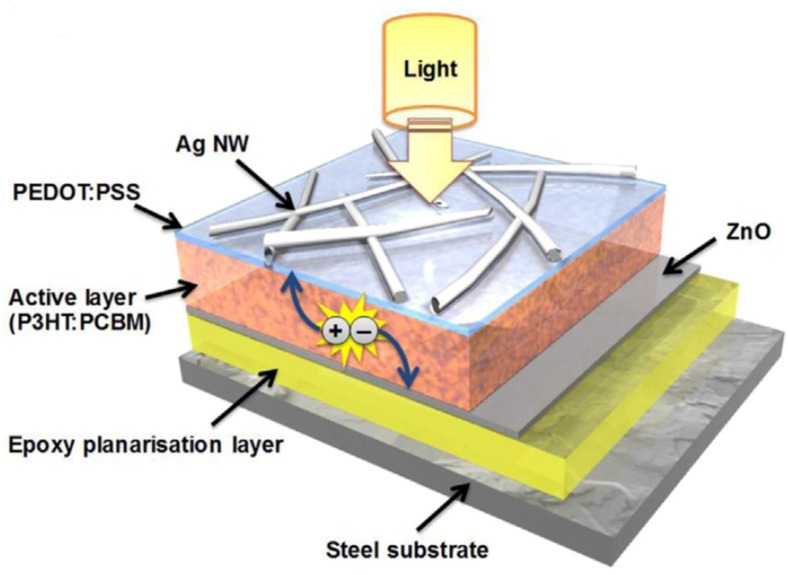
Structure of OPV device manufactured onto steel [101].

**Figure 26 materials-12-01731-f026:**
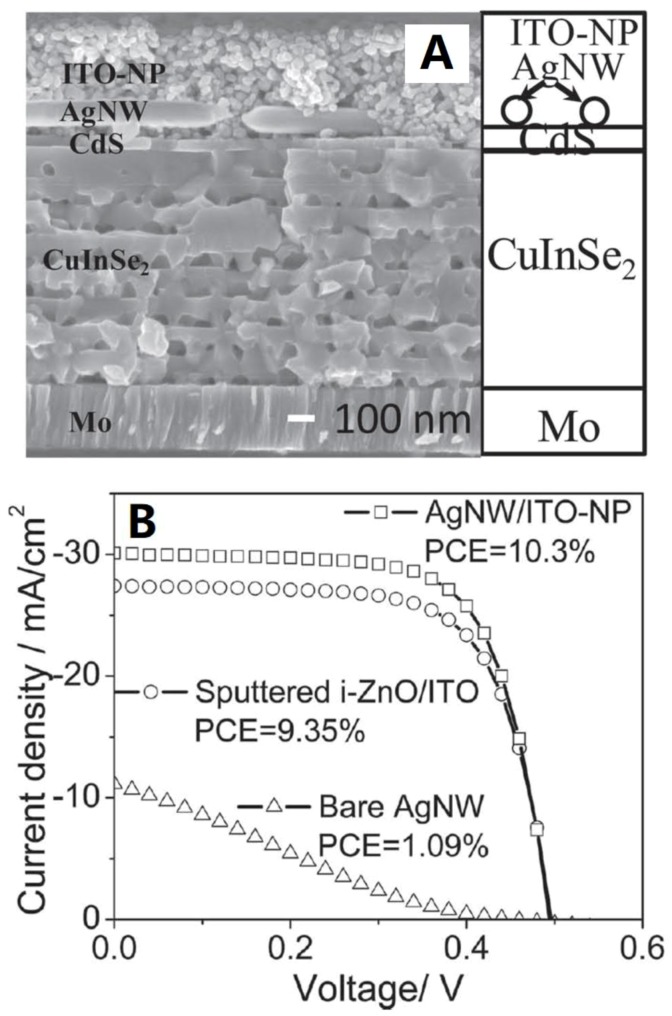
(**A**) Cross-sectional SEM image of the fully solution-processed CuInSe_2_ TCE with an AgNW/ITO-NP top contact and (**B**) J-V curves of the CuInSe_2_ TCEs using AgNW mesh, AgNW/ITO-NP film, and sputtered i-ZnO/ITO as top contacts [102].

**Table 1 materials-12-01731-t001:** Main feature, advantages and limitations of fabrication techniques of metallic NWs.

Approaches	Main Feature	Advantages	Limitations
Hard template method	The hard template serves as a powerful scaffold to control the diameter and length of NWs	The dimensions of metallic NWs can be easily shaped with the morphology of the used hard template	The purification process is time-consuming and tedious because of the fabrication and subsequent removal of template The removal of template may damage the structure of metallic NWs The remover is usually not environmental-friendly
Soft template method	The soft templates can dissolve in solution	The purification process is easy without the removal of template Metallic NWs can be scaled up readily	It is hard to control the morphology, size, and yield of metallic NWs It requires subsequent immobilization on matrices for many devices Precursors need to be carefully selected
Polyol method	Polyol is used as both solvent and reducing agent	It is able to produce AgNWs both at large scales and of high quality	The injection rate of EG solution needs to be accurately controlled in the self-seeding polyol process Capping agents may interact strongly with the metal surface The distribution of synthesized AgNWs is not easily controllable
Hydrothermal method	Water is used as solvent	Environmentally friendly, feasibility of large-scale fabrication of ultralong AgNWs and CuNWs Fewer constraints regarding precursor selection and the conditions of the solvents and reactions	Surface oxidation of metallic NWs may occur
Microwave method	Microware radiation is used as heating source	Fast and large-scale synthesis for metallic NWs Easy and highly reproducible process	The morphology and distribution of metallic NWs are very susceptible to changes with reaction parameters, such as microwave power, concentration of reducing agent, and surfactant and reaction time

**Table 2 materials-12-01731-t002:** Summary of metallic NWs for thermal conductivity enhancement of phase change material (PCM).

References	Author	Year	Potential Building Applications	Base PCM	NW	Performance
[14]	Shah et al.	2014	Passive heating and cooling Solar absorption cooling Evaporative cooling Radiative cooling Air conditioning system	CaCl_2_∙6H_2_O (T_m_ = 29 °C)	CuNWs	A highest CuNWs yield of 50% was obtained Doping 0.17 wt.% CuNWs resulted in 52% enhancement in the thermal conductivity of composite PCM
[77]	Zhu et al.	2014	Solar domestic hot water system Solar chimney Solar desalination Photovoltaic/thermal system	Palmitic acid (T_m_ = 64 °C)	CuNWs	Thermal conductivity of form-stable PCM was increased from 0.377 W m^−1^ K^−1^ to 0.455 W m^−1^ K^−1^ by doping 11.2 wt.% of CuNWs The latent heat of the CuNWs doped form-stable PCM could attain 149 J g^−1^
[64]	Zeng et al.	2012	Solar air heating system District heating system Waste heat recovery	Tetradecanol (T_m_ = 40 °C)	CuNWs	Thermal conductivity of PCM was increased from 0.32 W m^−1^ K^−1^ to 2.86 W m^−1^ K^−1^ by doping 11.9 vol.% of CuNWs The latent heat of the CuNWs doped composite PCM could attain 86.95 J g^−1^
[78]	Zeng et al.	2010	Solar air heating system District heating system Waste heat recovery	Tetradecanol (T_m_ = 40 °C)	AgNWs	Thermal conductivity of PCM was increased from 0.32 W m^−1^ K^−1^ to 1.46 W m^−1^ K^−1^ by doping 11.8 vol.% of AgNWs The latent heat of the AgNWs doped composite PCM could attain 76.5 J g^−1^
[79]	Deng et al.	2016	Solar domestic hot water system Solar chimney Solar desalination Photovoltaic/thermal system	Expanded vermiculite (T_m_ = 60 °C)	AgNWs	Thermal conductivity of PCM was increased from 0.25 W m^−1^ K^−1^ to 0.68 W m^−1^ K^−1^ by doping 19.3 wt.% of AgNWs The latent heat of the AgNWs doped composite PCM could attain 99.1–96.4 J g^−1^ The supercooling degree of composite PCM was reduced by 7 °C

**Table 3 materials-12-01731-t003:** Summary of AgNWs for improving the thermal transport of the base fluid.

References	Author	Year	Potential Building Applications	Base Fluid	Performance
[80]	Carbajal–Valdez et al.	2019	Air conditioning system including air handling unit, fan coil unit, cooling tower, chiller plant, etc. Solar energy harvesting system including solar panel, heat exchanger, water tank, etc.	DI water	Adding 1.74 × 10^−4^ vol.% AgNWs resulted in 20.8% enhancement in the thermal conductivity of nanofluid (from 0.613 W m^−1^ K^−1^ to 0.723 W m^−1^ K^−1^)
[81]	Zhang et al.	2017		EG	The thermal conductivity enhancement of 13.42% was achieved for the EG/AgNWs nanofluid by doping 0.46 vol.% of AgNWs (from 0.256 W m^−1^ K^−1^ to 0.284 W m^−1^ K^−1^)
[82]	Şimşek et al.	2018		PVP–DI water	A maximum enhancement of 56% has been observed in the heat transfer coefficient with no increase in the pumping power

**Table 4 materials-12-01731-t004:** Summary of AgNWs for fabricating TCE used in ECDs.

References	Author	Year	Potential Building Applications	Substrate	Performance
[87]	Lin et al.	2017	Electrochromic windows (ECW)	PET	Sheet resistance of 15 Ω sq^−1^ and transmittance of 95% were obtained for the TCE Switching time of coloration of 4.3 s and coloration efficiency of 120 cm^2^ C^−1^ were achieved for the ECW
[89]	Mallikarjuna and Kim	2018		PET	Sheet resistance of 0.714 Ω sq^−1^ and transmittance of 90.9% at wavelength of 550 nm were obtained for the TCE Switching response (12 s for coloration and 10 s for bleaching), coloration efficiency of 33.4 cm^2^ C^−1^ were obtained and device responsivity of 90% were achieved for the ECD
[90]	Kim et al.	2018		Graphene	Sheet resistance of 24–28 Ω sq^−1^ and transmittance of 58–67% were obtained for the TCE A maximum optical contrast of 15% at wavelength of 630 nm was achieved for the electrochromic film
[91]	Yuksel et al.	2017		PET	Very fast switching response of 0.86 s (coloration) and 0.57 s (bleaching) was obtained for the ECD
[92]	Zhou et al.	2017		PVA	High coloration efficiency of 50.0 and 86.9 cm^2^ C^−1^ at 630 and 1100 nm and switching response of 17.5 s for coloration and bleaching were obtained for the ECD

**Table 5 materials-12-01731-t005:** Summary of AgNWs and CuNWs for fabricating TCE used in Photovoltaic (PV) cells.

References	Author	Year	Potential Building Applications	PV Device Structure	Performance			
					Power Conversion Efficiency (PCE) (%)	Short-Circuit Current Density (J_SC_) (mA cm^−2^)	Fill Factor (FF) (%)	Open-Circuit Voltage (V_OC_) (V)
[97]	Kim et al.	2011	Building-integrated photovoltaic (BIPV)	Al/P3HT:PCBM:AgNWs/PEDOT:PSS/ITO/Glass	3.9	9.3	63.3	0.66
[98]	Lu et al.	2015		AgNWs/ PEDOT:PSS:MoO_3_/ P3HT:PCBM/ZnO/ITO/Glass	2.7	8.4	54.0	0.60
[99]	Maisch et al.	2016		AgNWs/PEDOT:PSS/PV2000:PC_70_BM/ZnO/AgNWs/Glass	4.3	10.7	52.8	0.76
[100]	Park et al.	2016		Al/NDP9/BF-DPB /BPAPF /BPAPF/DCV 5T-Me/C_60_/Bis-Hfl-NTCDI:W2(hpp)4/ PEDOT:PSS:AgNWs/Glass	7.1	12.3	60.6	0.95
[101]	Ding et al.	2016		AgNWs/PEDOT:PSS/P3HT:PCBM/Epoxy/Steel	2.3	8.2	53.0	0.53
[102]	Chung et al.	2012		ITO/AgNWs/CdS/CuInSe_2_/Mo/Glass	10.3	30.1	69.3	0.49
[103]	Singh et al.	2018		ZnO/AgNWs/CdS/CuInSe_2_/Mo/Glass	13.4	33.8	62.0	0.64
[104]	Teymouri et al.	2019		Al/P3HT:PCBM/PEDOT:PSS/PET:AgNWs	3.1	9.1	68.0	0.51
[105]	Wei et al.	2018		PEDOT:PSS:AgNWs/P3HT:PC_61_BM/ZnO/Al	3.3	9.9	55.7	0.59
[106]	Won et al.	2014		CuNW/AZO/i-ZnO/ZnS/CIGSSe/Mo/Glass	7.1	33.7	36.0	0.58

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
