# Peer review of "Multifunctional Metallic Nanowires in Advanced Building Applications"

_materials, 2019, doi:10.3390/ma12111731_

Reviewer 1 Report

Manuscript number: materials-486048

Title: Multifunctional Metallic Nanowires in Advanced Building Applications

In proposed manuscript, authors gathered literature information regarding silver and cooper high aspect ratio nanowires and their application in building and construction fields. Special focus is put on two chemical approaches for preparation of that kind of nanowires, namely polyol and hydrothermal methods. Despite the fact that the authors supplement their description with many SEM photographs taken from the literature, they do not compare these methods with each other and with other techniques of obtaining metallic nanowires (three publications on the templates-assisted synthesis are mentioned at the beginning of 2.1 section).

Section 3 is divided into subsections and there are literally listed general techniques for the characterization of nanomaterials such as SEM, TEM etc.

Also, in section 3.1 the authors identify thermoelectric properties with electrical and thermal conductivity of the material, which is an obvious mistake. From at least two decades, scientist developing this field presented results proving that not metallic but semiconducting nanowires possess desirable thermoelectric properties. What is more, illustrating thermoelectric properties of silver nanowires with just electric resistivity and thermal conductivity, completely neglecting Seebeck coefficient of this material is just a prove that the authors do not carry out sufficient literature research.

Section 3.3 contains 9 lines of text which can be summarized in one sentence: UV-Vis absorption of silver nanostructures depends on their shape and size. I would not call this an exhaustive description of the optical properties of metallic nanowires (the authors did not even mention Cu NWs).

Section 4 is the most extensive fragment of the presented manuscript. The authors mention several applications of nanowires, which in the future can be used in the construction industry. In my opinion, this is the most valuable part of this work. Every subsection is summarized in table. However, when analyzing these tables, one can conclude that one more time authors focus on the parameters of synthesis and the obtained geometric dimensions of nanowires, not their performance. Of all the subsections, the part concerning the electrochromic windows is drawn most carefully, and provides all necessary information. In my opinion, the whole manuscript should be kept in such quality. In this subsection authors refer to figures 20-A and 20-B, which I believe should be 21A and B. In the next subsection (page 28) authors refer to figure 24 which is not present in this manuscript. Also, in case of usage of Ag NWs in photovoltaic cells authors do not exhaustively describe the topic. One just need to google "silver nanowire photovoltaic cell" and many publications from the last 3 years are displayed, but authors cite only 6 from 2010-2016.

Taking into account the above remarks, I would not recommend this manuscript for publication.

Author Response

In proposed manuscript, authors gathered literature information regarding silver and cooper high aspect ratio nanowires and their application in building and construction fields. Special focus is put on two chemical approaches for preparation of that kind of nanowires, namely polyol and hydrothermal methods.

Point 1: Despite the fact that the authors supplement their description with many SEM photographs taken from the literature, they do not compare these methods with each other and with other techniques of obtaining metallic nanowires (three publications on the templates-assisted synthesis are mentioned at the beginning of 2.1 section).

Response 1: Thank you for your comment. We have compared the different synthetic techniques of obtaining metallic nanowires by adding a new table (Table 1) in Page 9 Line 188. The techniques include hard template method, soft template method, polyol method, hydrothermal method and microwave radiation method. We have summarized the main feature, advantages and limitations these methods in Table 1.

Point 2: Section 3 is divided into subsections and there are literally listed general techniques for the characterization of nanomaterials such as SEM, TEM etc. Also, in section 3.1 the authors identify thermoelectric properties with electrical and thermal conductivity of the material, which is an obvious mistake. From at least two decades, scientist developing this field presented results proving that not metallic but semiconducting nanowires possess desirable thermoelectric properties. What is more, illustrating thermoelectric properties of silver nanowires with just electric resistivity and thermal conductivity, completely neglecting Seebeck coefficient of this material is just a prove that the authors do not carry out sufficient literature research.

Response 2: Thank you for reminding us the obvious mistake we made regarding the thermoelectric properties. In Page 11 Line 209, we have changed the title of section 3.2 to “Electrical and thermal conductivity”. We have no intention to discuss the thermoelectric properties of metallic NWs. We focused on two factors (size effect and temperature) affecting the electrical and thermal conductivity of metallic NWs.

Point 3: Section 3.3 contains 9 lines of text which can be summarized in one sentence: UV-Vis absorption of silver nanostructures depends on their shape and size. I would not call this an exhaustive description of the optical properties of metallic nanowires (the authors did not even mention Cu NWs).

Response 3: Thank you for your comment. In Page 11 Line 233, we have discussed the Plasmonic effect of metals and the surface plasmon resonance (SPR) behavior and the methods to examine the SPR behavior. In addition, the UV-Vis absorption peak of CuNWs has been supplemented (Page 12 Line 247).

Point 4: The authors mention several applications of nanowires, which in the future can be used in the construction industry. In my opinion, this is the most valuable part of this work. Every subsection is summarized in table. However, when analyzing these tables, one can conclude that one more time authors focus on the parameters of synthesis and the obtained geometric dimensions of nanowires, not their performance. Of all the subsections, the part concerning the electrochromic windows is drawn most carefully, and provides all necessary information. In my opinion, the whole manuscript should be kept in such quality.

Response 4: Thank you for your comment. We have supplemented more information of TES (Page 12 Line 262). But to the best of our knowledge, the performance of NEPCM and nanofluid for improving building energy efficiency is still lacking in the published literature. We have added this as an outlook in the section 5 (Page 36 Line 587).

Point 5: In this subsection authors refer to figures 20-A and 20-B, which I believe should be 21A and B. In the next subsection (page 28) authors refer to figure 24 which is not present in this manuscript.

Response 5: Thank you for this excellent observation. We have rectified these figure number mistakes.

Point 6: Also, in case of usage of Ag NWs in photovoltaic cells authors do not exhaustively describe the topic. One just need to google "silver nanowire photovoltaic cell" and many publications from the last 3 years are displayed, but authors cite only 6 from 2010-2016.Taking into account the above remarks, I would not recommend this manuscript for publication.

Response 6: Thank you for comment. In section 4.4 (Page 30 Line 462), we have supplemented 6 more papers [98-100] and [102-104] to support the recent research of using silver nanowires for photovoltaic cells. We have also rectified the Table 5 (Page 35 Line 556) to sort out the PV device and their performance.

Reviewer 2 Report

The paper review the research and progress regarding multifunctional metallic NWs and their profound building applications.The referee recommends this manuscript for being published with the minor revision. To improve the scientific quality and readability of the manuscript, the referee suggested the following comments:

The capital letter of graph items should be consistent between the figures (e.g Fig.4, Fig.6 a,b,c,...) and figure captions(A,B,C...).  

Parts of the reference did not contain pages information (E.g reference 9, 38, 59, 61...).

Author Response

The paper review the research and progress regarding multifunctional metallic NWs and their profound building applications. The referee recommends this manuscript for being published with the minor revision. To improve the scientific quality and readability of the manuscript, the referee suggested the following comments:

Point 1: The capital letter of graph items should be consistent between the figures (e.g Fig.4, Fig.6 a,b,c,...) and figure captions(A,B,C...). 

Response 1: Thank you for this excellent observation. We have rectified these figure problems by using uniform caption.

Point 2: Parts of the reference did not contain pages information (E.g reference 9, 38, 59, 61...).

Response 1: Thank you for this excellent observation. We have double checked and rectified these reference page problems.

Reviewer 3 Report

The authors of the article with title “Multifunctional Metallic Nanowires in Advanced
Building Applications” present deep review of the main properties, main methods for deposition, main characterization methods and main developed application of metallic nanowires. The study is written using simple and understandable style. However, the significance of the study is not a compromise with the quality of this work.
It was presented two methods for deposition of AgNW and CuNW- polyol and hydro-thermal method. The advantages and disadvantages of the methods as well was presented.  
It was discussed only AgNW and CuNW as a most well developed nanowires and materials incorporated in many applications like thermal energy storage (TES), electrochromic windows (ECW), transparent conductive electrodes (TCE), phase change materials (PCM) and photovoltaic (PV) cells. Every presented application of metallic nanowires is followed by adequate tables. The presented figures are clear, understandable and enough for first stage of understanding the author’s ideas.
The quality of the paper is high enough and is suitable for publication as it is. The minor corrections must be applied.

Author Response

The authors of the article with title “Multifunctional Metallic Nanowires in Advanced

Building Applications” present deep review of the main properties, main methods for deposition, main characterization methods and main developed application of metallic nanowires. The study is written using simple and understandable style. However, the significance of the study is not a compromise with the quality of this work.

It was presented two methods for deposition of AgNW and CuNW- polyol and hydro-thermal method. The advantages and disadvantages of the methods as well was presented. It was discussed only AgNW and CuNW as a most well developed nanowires and materials incorporated in many applications like thermal energy storage (TES), electrochromic windows (ECW), transparent conductive electrodes (TCE), phase change materials (PCM) and photovoltaic (PV) cells. Every presented application of metallic nanowires is followed by adequate tables. The presented figures are clear, understandable and enough for first stage of understanding the author’s ideas.

Point 1: The quality of the paper is high enough and is suitable for publication as it is. The minor corrections must be applied.

Response 1: Thank you for comment. We have carried out several minor corrections (marked with red), both technically and grammatically to improve the readability of this review paper.

Round  2

Reviewer 1 Report

Manuscript number: materials-486048-v2

Title: Multifunctional Metallic Nanowires in Advanced Building Applications

In review  version of the manuscript, the authors have improved the quality of the text, completed the missing literature, refined the information in the tables. The quality and completeness of the presented issues has also been improved. In this version, the manuscript is written legibly, clearly and logically. In addition, beside few typos, the text is written carefully.

I recommend that the manuscript should be accepted for publication